# Epidemiology and Trends over Time of Foreign Body Injuries in the Pediatric Emergency Department

**DOI:** 10.3390/children8100938

**Published:** 2021-10-19

**Authors:** Honoria Ocagli, Danila Azzolina, Silvia Bressan, Daniele Bottigliengo, Elisabetta Settin, Giulia Lorenzoni, Dario Gregori, Liviana Da Dalt

**Affiliations:** 1Unit of Biostatistics, Epidemiology and Public Health, Department of Cardiac, Thoracic, Vascular Sciences, and Public Health, University of Padova, via Loredan 18, 35128 Padova, Italy; honoria.ocagli@unipd.it (H.O.); danila.azzolina@unipd.it (D.A.); Daniele.Bottigliengo@ubep.unipd.it (D.B.); elisabetta.settin@ubep.unipd.it (E.S.); giulia.lorenzoni@unipd.it (G.L.); dario.gregori@unipd.it (D.G.); 2Department of Medical Science, University of Ferrara, via Fossato Mortara 64 b, 44121 Ferrara, Italy; 3Pediatric Emergency Unit—Department of Women’s and Children’s Health, University of Padova, via Nicolò Giustiniani, 3, 35128 Padova, Italy; silvia.bressan@aopd.veneto.it

**Keywords:** foreign body, pediatric emergency department, children, injury diagnosis, child, descriptive epidemiology

## Abstract

This paper presents the epidemiology of foreign body injuries in the Pediatric Emergency Department (PED) of Padova (Italy) along with its trends over an eleven-year period based on administrative data. Annual incidence rates (IRs) of PED presentations for foreign body (FB) injuries per 1000 person-years were calculated. Univariable and multivariable generalized linear (GLM) Poisson models were estimated to evaluate the relationship between FB injury incidence and year, triage priority, nationality, injury site, and FB type. During the study period, there were 217,900 presentations of pediatric residents in the province of Padova; of these, 3084 (1.5%) reported FB injuries involving the ears, nose, throat, gastrointestinal tract or eyes. The annual IR of FB injury episodes increased from 10.45 for 1000 residents in 2007 (95% CI, 9.24, 11.77) to 12.66 for 1000 residents in 2018 (95% CI, 11.35, 14.08). Nonfood items were the FBs that were most frequently reported. The intermediate urgent triage code was the most represented for FB injuries, with IRs ranging from 5.44 (95% CI: 4.59, 6.40) in 2008 to 8.56 in 2018 (95% CI: 7.50, 9.74). A total of 170 patients who presented for FB injuries were hospitalized (5.5%). The annual FB-related injury IR has increased over time, although most episodes are not life threatening. Educational and prevention programs on FB-related injuries should be promoted and dedicated to childcare providers.

## 1. Introduction

Injuries related to foreign bodies (FBs) occur frequently in children, particularly in those younger than three years of age [1]. FB-related injuries may cause significant morbidity and can occasionally lead to a fatal outcome. In Italy, the estimated incidence of FB injuries in 2004 was reported to be approximately 0.2% [2]. The same study [2] estimated, with the scale-up approach, that 15,829 children experienced an FB injury in 2004 and 12,844 (81%) were hospitalized. Italy is one of the countries participating in the Surveillance System on Foreign Body Injuries in Children (Susy Safe) project [3,4], an international surveillance registry for injuries in children between 0 and 14 years of age. The registry provides a risk-analysis profile for products causing injury; it evaluates the impact of socioeconomic disparities and involves consumer associations. Along with government institutions, this system has contributed to creating guidelines for food injury prevention in children [5]. The Susy Safe project provides the possibility to compare single center data with a broader dataset, as in the study of Aydin et al. [6], when comparing FB injury patterns between Turkey and European countries. The registry is also useful to detect patterns of FB injuries across different countries, as done when comparing Bosnia Herzegovina [7] and India [8] to European countries.

FB injuries can involve different anatomical sites [9], such as the aerodigestive tract [10], the eyes [11] and, less frequently, the genitals and rectum [12]. The objects responsible for injuries can be of different shapes and natures [9], such as organic foreign bodies (i.e., food items) [13] or magnetic foreign bodies [14]. The European Survey on Foreign Bodies Injuries (ESFBI) reported complications in 70 out of 553 (12.7%) cases involving the upper airways [15], in 65 out of 443 (13%) cases involving the external auditory canal [16] and in 59 out of 688 (8.6%) cases involving the nose [17]. Choking injuries related to inhalation of FBs are those that more frequently result in death compared to FB-related injuries in other sites [6,7,8]. In 2000, FB aspiration had an incidence of 29.9 in 100,000 children and led to death in 160 children in the US [18].

Given the relevant incidence of the problem and the possible complications depending on the type of FB, both prevention and timely recognition are important to improve patient outcomes [19]. Unfortunately, delays in recognition of FB-related injuries still occur due to insufficient time to understand the history behind a critical presentation, scant information on the event, which may not be witnessed by a supervising adult, [20] or the aspecific nature of symptoms at presentation [21].

Knowing what kind of objects are involved in FB injuries could help best define the interventions that are needed to prevent these injuries. This gap possibly contributes to the low awareness of the problem among caregivers and childcare providers.

Several studies, generally single center studies [22,23], have described FB injuries; however, they focused on hospitalized patients [22], who represent the most severe end of the spectrum. Investigating the frequency and outcome of FB-related injuries in the acute care setting has the advantage of providing data on the broader spectrum of FB-related injuries in children, adding information to the limited existing epidemiological knowledge on the topic [23].

While the epidemiology of choking injuries has been well described [24], scant information is available on FB-related injuries involving other body sites [22,23].

This study aimed to assess the incidence rate (IR) of FB-related injuries in children presenting in the acute care setting of a tertiary-care pediatric emergency department (PED) in northeastern Italy to describe the epidemiology of these injuries across the severity spectrum and type of injury site. We did not have a predefined hypothesis for the primary aim, as this was an exploratory study, though we expected FB-related injury presentations to be in line with national and international trends in terms of the types and sites of FB injuries.

As a secondary aim, we set out to describe the characteristics of FB-related injury presentations compared with non-FB-related injury presentations and to evaluate the trend of the phenomenon over time.

## 2. Materials and Methods

### 2.1. Study Design

This was a retrospective study performed over an 11-year period of time using the electronic PED database of the Padova University Hospital, located in the Veneto region of Italy.

### 2.2. Setting

The PED provides primary and secondary care for a metropolitan area of 350,000 people (45,000 people younger than 15 years) and tertiary care for a regional and extra regional population, with approximately 25,000 PED visits per year.

Children presenting to the PED are seen based on the triage code they receive after the initial assessment of a triage nurse. The triage priority level during the study years was established following a local triage protocol, which classified children as critical—the color red (requiring an immediate medical assessment), very urgent—the color yellow (requiring an assessment within 30 min), urgent—the color green (requiring an assessment within an hour) and nonurgent-color white [25].

### 2.3. Inclusion and Exclusion Criteria

The electronic records of children aged between 0 and 15 years living in the province of Padova who presented to the PED for a FB-related injury to the ears, nose, throat, or the respiratory or gastrointestinal tracts based on the discharge diagnosis were included in the study (flowchart, Figure 1). Children who had FB injuries to other sites of the body, such as the skin or feet, or having injuries related to medications, caustics or chemicals, were excluded. The analysis was carried out only on the child residents in Padova, identified by selecting the Italian health care district reference codes (ASL) 050116 and 050506.

The words “obstruction”, “inhaled”, “inhalation”, “choking”, “inserted”, “swallowed” and “foreign” were searched in the “diagnosis” and “chief complaint” fields to select the records meeting the inclusion criteria. This search strategy resulted in higher sensitivity than the International Classification of Disease strategy for retrieving information on the anatomical location of FB injuries.

### 2.4. Variables’ Characteristics

Data concerning patient characteristics (age, gender, nationality), FB type (food or not food), body site involved (mouth-digestive tract, respiratory tract, nose-eyes-ears d genitourinary tract), discharge/admission and triage priority code were collected.

Children were divided according to their developmental stages as follows: (i) infant-toddler (<2 years old); (ii) preschooler (<6 years old); and (iii) schooler (>6 years old).

FBs were classified into food or nonfood types according to the description retrieved in the database.

### 2.5. Statistical Analysis

During the study period (2007–2018), 217,900 presentations of children residing in the Padova municipality were recorded in the PED hospital database. Among these, 3217 (1.5%) records were identified as FB injury events meeting the study inclusion criteria (flowchart, Figure 1).

Descriptive statistics of the data according to the presence of FBs were performed. Data are reported as medians, I, and III quartiles as a measure of dispersion for continuous variables and percentages for categorical variables. The Wilcoxon test was used to compare continuous variables; the likelihood ratio chi-square test from the proportional odds model was carried out to compare categorical ordered variables; and the Pearson chi-square test was used for categorical nonordered variables [26]. The logistic regression univariable odds ratios (ORs) are reported in the descriptive table for the FB injury outcomes, together with the 95% confidence intervals (CIs). The OR referring to an interquartile effect is shown for the continuous variable.

### 2.6. Outcome Measurements

Our primary outcome was the incidence ratio (IR), which is defined as the occurrence of new cases in a population over a period of time and was used to estimate the occurrence of FB injuries in the reference population. The IR was stratified according to sex, nationality, FB type, triage priority category, and reference year. The IR denominator, consisting of the Padua resident population, was determined by consulting the official Italian National Census and intercensus data (2007–2018) provided by the National Institute of Statistics ISTAT (www.istat.it, accessed on 27 September 2019) in people under 15 years of age. The absolute number of FB injury events has been reported with a 95% IR Poisson confidence interval.

Our secondary outcomes were the characteristics of FB-related injury presentations in terms of FB type, injury site and severity.

A univariable and multivariable GLM Poisson model [27] was estimated to evaluate the relationship between the FB IR and year, triage priority category, nationality, injury site, and type of FB. The nonlinear age effect was modeled considering a restricted cubic spline approach (3 knots). Statistical analysis was performed with the R System [28] and rms package 26. Significance was assessed for a *p* value < 0.05.

## 3. Results

The characteristics of the overall population according to the presence/absence of FBs are reported in Table 1. Children in the FB injuries group were younger than the children in the no-FB injuries group; 1112 (36%) FB injuries occurred among infants-toddlers and 980 (32%) occurred among preschoolers in the first group and 80,043 (38%) injuries and 46,980 (23%) injuries occurred among infants-toddlers and preschoolers, respectively, in the no FB group (*p* < 0.001). Among males, there were 122,794 (56%) FB injuries in the infant-toddler group and 1790 (55%) in the preschoolers group (*p* = 0.265). Regarding nationality, non-Italian ancestry were most frequently reported in the no FB injury group (46,173, 22%) compared with 617 in the FB group (20%) (*p* = 0.005). Appendix A reports the descriptive characteristics of FB characteristics in FB-related presentations according to age. The leading cause of FB injuries was nonfood items, especially for children less than 6 years old. FB injuries mainly involved the ears, nose and throat for children less than six years old; moreover, pediatric patients admitted to the hospital were predominantly toddlers (Appendix A).

The most represented triage category overall was the intermediate urgency level (green color) in both groups, 140,244 (67%) and 2046 (66%) in the non-FB and FB groups, respectively. A higher proportion of the urgent triage category (yellow color) was observed in the FB injury group (613 cases, 20% compared with 35,368 cases, 17%, *p* < 0.001). The very urgent triage priority category (red color), representing a critical condition, was assigned to 27 children (1%) with FB-related injury presentations. Of the 3217 presentations for FB injuries, 2566 (83.20%) cases had information on the anatomical location involved and 591 (19.16%) had information on the type of FB.

The ears, nose and throat were the sites most often involved (1169, 46%) in patients with FB injuries. Table 1 also reports the odds ratios for FB presentations. Preschoolers had a 1.5-fold higher OR (95% CI 1.38, 1.64) of having an FB injury than infants and toddlers.

In our study, hospitalization was necessary in only 170 (6%) cases of FB-related injuries (147 among infants and toddlers, 61 among preschoolers and 56 among schoolers). Hospitalized children were younger than discharged children; in the first group, there were 94 infants-toddlers (55%) with FB injuries compared with 1007 (35%) in the second group (*p* < 0.001). Children with ear-nose-throat injuries and yellow and red triage colors required more hospitalization (*p* < 0.001) (Table 2).

### 3.1. The Trends over Time of Foreign Body Injuries

FB injuries were summarized into yearly counts to evaluate their trends over time. The IR according to the time of the overall cases is reported in Table 3. Appendix A reports yearly IRs divided by gender, nationality, FB type, triage priority category, and body site involved. Significant differences in the FB IR of the resident population between males and females were identified, with a 95% CI of 8.95 (8.49, 9.42) and a 95% CI of 7.92 (7.47, 8.38), respectively (*p* = 0.002) (Appendix A). The yearly IR of FB injury episodes ranged from 10.45 for 1000 residents in 2007 (95% CI 9.24, 11.77) to 12.66 in 2018 (95% CI 11.35, 14.08), which was 254 and 321 episodes per 100,000 person-years, respectively. Among the residents, those with an Italian nationality were slightly overrepresented in the group with an FB injury except for three years. For the FB type, no-food items were the most represented in each year. For severity, the green color triage code was the most represented every year, with IR ranging from 6.67 (95% CI 5.72, 7.73) in 2007 to 8.56 in 2018 (95% CI 7.50, 9.74) (Table 3).

### 3.2. Univariable and Multivariable Models

The univariable models (Figure 2) showed that the IR was higher in male children (log (IRR) 0.154, *p* < 0.001), in no-food FBs (log (IRR) −0.693, *p* < 0.001) and in green triage color codes (log (IRR) 1.637, *p* < 0.001).

These results were confirmed by the multivariable model (Figure 3). Lower age and the male sex had a great impact on the IR in the multivariable model; moreover, the IR increased over the years. The nonlinear shape of the predicted age effect revealed that the FB injury IR rises in the first three years of age and then decreases in older children (Figure 3).

## 4. Discussion

This work describes the characteristics of FB-related injuries and outcomes in children under the age of 15 years from 2007 to 2018 at the University Hospital of Padova. The main findings are the following: (1) the trend in the FB IR presentations is still increasing; (2) boys had more FB injuries than girls; (3) the gastrointestinal tract and respiratory tract were the most frequent sites of injury; (4) FB injury episodes were mainly reported in younger children; and (5) food-related episodes were the most frequently reported FB-related injuries.

The increasing trend in presentations for FB-related injuries over the years highlights the need to enhance injury prevention measures at the community level. This happens despite the effort to increase recognition by health care professionals and public awareness on the topic of FB injuries in children, for example, with the publication of the National Guidelines for Preventing Choking Injuries [5]. Few studies have reported the trends and characteristics of pediatric FB-related injuries. In the Italian territory, for example, trends related to FB ingestions that required upper endoscopy have been reported [29]. The increase we found was lower than that reported by Park et al. [30] in his study in a PED in South Korea. His study described an increase in cases from 2.15 in 2010 to 4.36 in 2014 per 1000 population (*p* < 0.001) in four years [30]. Sinikumpo [22], in his study in a pediatric trauma unit in Finland, reported no change in the incidence rate of FB injuries (*p* = 0.634) during a five-year period between 2008 and 2013. These differences were probably related to the different pediatric age groups that were analyzed. Park et al. [30] reported injury-related ED visits among children younger than seven years of age using the National Emergency Department Information System of South Korea as their source of data. The Finnish study included all children <16 years of age. The number of cases reported in our research is much higher than the 455 cases reported in a Romanian study [31]. This study was conducted on a five-year period of ED presentations involving only events that required access to the ear-nose-throat department.

The incidence of these episodes is usually studied in terms of hospitalization, even if the ED is the first service that provides care for these patients. In our study, only a small proportion of children presenting to the ED required hospital admission for the management of FB injuries (170, 5.5%). This percentage is lower than the results of a survey conducted among Italian mothers, where one out of eight injuries in children presenting to an emergency department resulted in hospitalization [2]. The survey applied the scale-up technique to estimate the number of injuries. The questionnaire was completed by 1081 women aged 18–50 years, and they recalled 437 children who suffered a foreign body-related injury [2]. Moreover, in a recent Italian study over the 2001–2013 period, the number of hospitalizations decreased [24]: this may be explained by the higher attention of policy-makers on the theme, proven, for example, by the introduction of the Toy Safety Directive 2009/48/EC in European countries and the subsequent publication of the guidelines for preventing choking injuries [5]. 

The other findings of this study are consistent with the results of another study conducted in an ED setting in South Korea [30]. For example, males reported more episodes of FB injuries, a result coherent with most of the literature on the topic. Only two studies found that females were the most involved in these kinds of injuries [23,31].

Regarding the site of the injury, the gastrointestinal tract [32] was the site that was most involved. However, the respiratory tract was the second anatomical site that was most involved, in contrast with the results of previous studies [22,23,30,31], where the respiratory tract was the anatomical site most often involved in FB injuries. As already evidenced in other works, FB injuries occur prevalently in infants aged less than three years [22,23,33] because they try to explore objects using, in particular, their mouth [1,21].

Although data were limited on the nature of the FB, food items were the most frequently reported. This is not in line with the results of a recent study that reported a 10-year experience in managing pediatric foreign body ingestions in an emergency clinic of an Italian tertiary hospital [29]. Conversely, previous similar work has shown that food items are the objects prevalently reported in FB injuries [30,31]. The prevalence of food items causing FB injuries in the literature has suggested the need for further work in prevention policy implementation and surveillance. These interventions are oriented toward parents and childcare providers [34]. The European Survey on Foreign Body Injuries (ESFBI) addresses this problem. Some EU countries, in agreement with the European Directive, provide national guidelines tailored to choking injury prevention programs. The Italian [5] and European recommendations [35] for the prevention of foreign body inhalation are some examples, the first for food and the second for toys. In this direction, the Susy Safe registry is a useful instrument for injury surveillance purposes [3]. Therefore, the challenge is how to extend policies and guidelines to the community (i.e., parents, caregivers, teachers) and how to effectively translate the message of effective injury prevention strategies to the community.

Injury epidemiological knowledge would be a starting point to create focused prevention strategies, such as preventive programs and education policies oriented to parents and childcare providers [36,37]. For example, in Italy, the CHOP community interventional trial [38,39] has evaluated the effects of an educational prevention program dedicated to parents and childcare providers for preventing food choking injuries and increasing nutrition knowledge. The results of this trial have shown that a school-based public health intervention mediated by teachers is effective in increasing the knowledge of families on food choking injuries [40].

Our study, using administrative data from a single center, could not capture detailed information on the type of FB objects involved or on the presence/absence of parents, which is instead included in the more comprehensive Susy Safe registry [4]. However, we were able to collect exhaustive information on all ED visits for FB-related injuries and thus properly quantify the epidemiological phenomenon, as well as the burden of FB-related injuries for the ED service. Our results, despite the lower urgency of the triage level and the low percentage of cases resulting in hospital admissions, show that additional efforts are necessary to define injury prevention initiatives and interventions among childcare providers to reduce the occurrence of FB-related injuries. Moreover, our data will allow us to orient the policies both in the field of prevention of FB-related injuries in the community according to the age groups of the patient and the field of emergency service management. The potential role of the ED with respect to injury prevention should not be overlooked.

## 5. Study Limitations

Only ED injury cases were considered for the analysis; this aspect may involve an underestimation of the phenomenon because most of the episodes were self-resolved at home.

Another limitation to consider is the absence of a centralized encoding system (i.e., the ICD codes) of foreign body injuries in our data. Furthermore, information on the characteristics of the objects, the sites involved and the places where the events occurred are not exhaustive, since our study is based on administrative data.

Our study is limited by its retrospective single-center design, and our results cannot be generalized at the national level. The strength of our study relies on the fact that it shows a long period of IR trends and explores data that, especially in this case, are related to hospitalized episodes.

## 6. Conclusions

FB injury IR presentations at the PED are still increasing despite a growing awareness of policy-makers on the topic. However, FB episodes that required hospitalization are reduced compared to previous data in the national territory. Moreover, most of the episodes at triage are coded as not life-threatening. Strategies to increase knowledge on FB injuries should be implemented in our territory, and surveillance over time is useful to understand when is necessary to reinforce these strategies.

## Figures and Tables

**Figure 1 children-08-00938-f001:**
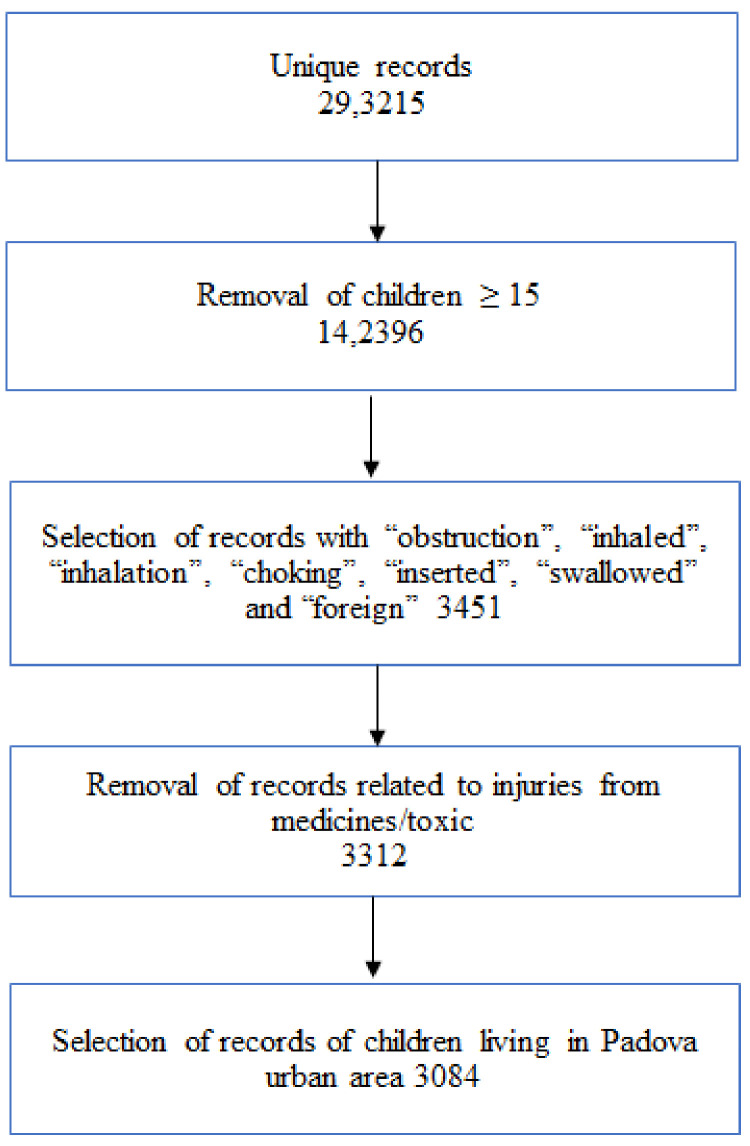
Flowchart of the study.

**Figure 2 children-08-00938-f002:**
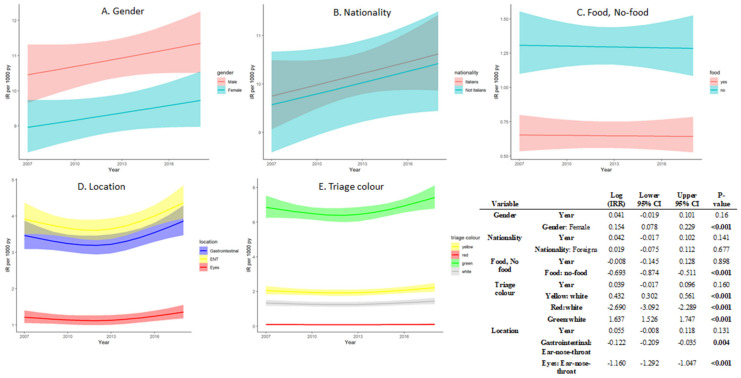
Poisson GLM univariable models according to the year of admission to the ED adjusted separately for sex, nationality, foreign body type, and anatomical site involved. The models estimated IRs, with 95% confidence bounds, over 1000 person-years, are represented on the plots. Model estimated log incidence rate ratio (IRR) is also reported in the tables for each model with standard errors (SE) and 95% confidence intervals.

**Figure 3 children-08-00938-f003:**
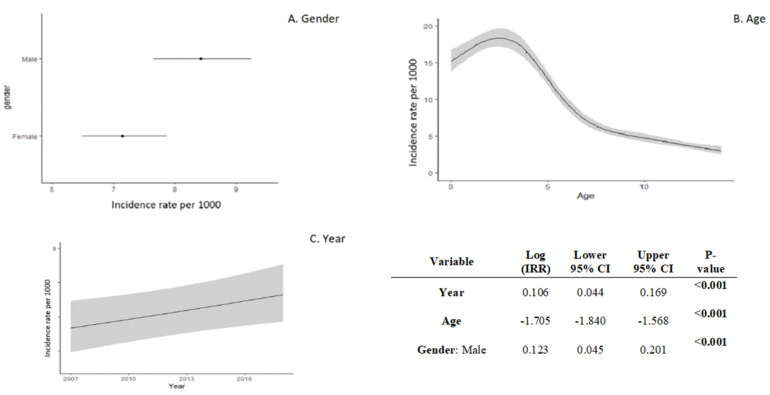
Poisson GLM multivariable models according to the year of admission to the ED adjusted for sex and age. The model-estimated IRs, with 95% confidence bounds over 1000 person-years, are represented on the plots. Model estimated log incidence rate ratio (IRR) is also reported in the table with standard errors (SE) and 95% confidence intervals. A restricted cubic spline (three knots) was considered to estimate the age effect.

**Table 1 children-08-00938-t001:** Descriptive characteristics of the population that had access to the pediatric emergency department of Padova. Continuous data are reported as medians (quartiles I and III), and categorical data are reported as absolute numbers (percentages). The logistic regression univariable odds ratio (OR) was reported for the foreign body presentation group. Interquartile effects are shown for the continuous variable.

Variable	Level	N	Other ED Presentations	FB-Related Presentations	FB Presentations Odds Ratio	Overall	*p* Value
			(N = 208,504)	(N = 3084)		(N = 211,588)	
Age (continuous)		211,646	1/4/9	2/4/7	0.79 [0.74–0.85]	1/4/9	0.003
Age categories	Infant-toddler (<2 y)	211,646	38% (80,043)	36% (1112)	*	38% (81,155)	<0.001
	Preschooler (<6 y)		23% (46,980)	32% (980)	1.5 [1.38–1.64]	23% (47,960)	
	Schooler (>6 y)		39% (81,481)	32% (992)	0.88 [0.8–0.96]	39% (82,473)	
Sex	Female	211,646	44% (90,870)	45% (1375)	*	44% (92,245)	0.265
	Male		56% (117,634)	55% (709)	0.96 [0.89–1.03]	56% (119,343)	
Nationality	Not Italian	211,646	22% (46,173)	20% (617)		22% (46,790)	0.005
	Italian		78% (162,331)	80% (2467)	1.14 [1.04–1.24]	78% (164,798)	
Outcome	Other	211,645	1% (1513)	1% (37)	*	1% (1550)	0.023
	Death		0% (9)	0% (0)	Not estimable	0% (9)	
	Discharged home		94% (195,569)	93% (2877)	0.6 [0.43–0.84]	94% (198,446)	
	Admitted		5% (11,412)	6% (170)	0.61 [0.43–0.87]	5% (11,582)	
Triage color	White	211,630	15% (31,201)	13% (398)	*	15% (31,599)	<0.001
	Green		67% (140,244)	66% (2046)	1.14 [1.03–1.27]	67% (142,290)	
	Yellow		17% (35,368)	20% (613)	1.36 [1.2–1.54]	17% (35,981)	
	Red		1% (1691)	1% (27)	1.25 [0.85–1.85]	1% (1718)	0.263

Abbreviations: ED, emergency department; FB, foreign body; * Reference value for odds ratio interpretation.

**Table 2 children-08-00938-t002:** Characteristics of the patients according to the outcome. Included in the category “other” are the patients that refused admission, failed to call, or voluntarily left the hospital. Continuous data are reported as medians (quartiles I and III), and categorical data are reported as absolute numbers (percentages).

Variable	Level	N	Discharge	Admission to Hospital	Other	Overall	*p* Value
			(N = 2877)	(N = 170)	(N = 37)	(N = 3084)	
Age (continuous)		3084	2/4/7	1/2/5	2/4/7	02/04/2007	<0.001
Age categories	Infant-toddler (<2 y)	3084	35% (1007)	55% (94)	30% (11)	36% (1112)	<0.001
	Preschooler(<6 y)		32% (927)	22% (38)	41% (15)	32% (980)	
	Schooler(>6 y)		33% (943)	22% (38)	30% (11)	32% (992)	
Sex	Female	3084	45% (1296)	38% (65)	38% (14)	45% (1375)	0.157
	Male		55% (1581)	62% (105)	62% (23)	55% (1709)	
Nationality	Not Italian	3084	20% (571)	21% (35)	30% (11)	20% (617)	0.322
	Italian		80% (2306)	79% (135)	70% (26)	80% (2467)	
Foreign body type	No-food	591	66% (365)	71% (27)	100% (2)	67% (394)	0.503
	Food		34% (186)	29% (11)	0% (0)	33% (197)	
Foreign body site	Ears, nose, throat	2566	44% (1061)	65% (100)	42% (8)	46% (1169)	<0.001
	Gastrointestinal tract		41% (974)	35% (54)	32% (6)	40% (1034)	
	Eyes		15% (357)	1% (1)	26% (5)	14% (363)	
Triage color	White	3084	13% (382)	6% (10)	16% (6)	13% (398)	<0.001
	Green		68% (1958)	38% (65)	62% (23)	66% (2046)	
	Yellow		18% (524)	48% (81)	22% (8)	20% (613)	
	Red		0% (13)	8% (14)	0% (0)	1% (27)	

**Table 3 children-08-00938-t003:** Incidence rate (IR) table of overall cases admitted to the emergency department of Padova according to year. The absolute number of FB injury events has been reported with a 95% Poisson confidence interval for IR over 1000 residents. Percentages (%) of FB injury events for PED presentations have also been reported with 95% confidence intervals. Univariate GLM Poisson *p* values for the relationship between time and FB incidence are indicated.

Year	2007	2008	2009	2010	2011	2012	2013	2014	2015	2016	2017	2018	*p* Value
Overall													
FB Events	254	210	250	278	283	260	227	248	238	239	276	321	
IR	10.45 (9.24, 11.77)	8.59( 7.51, 9.79)	10.04 (8.87, 11.32)	11.04 (9.82, 12.37)	11.2 (9.97, 12.54)	10.29 (9.11, 11.58)	8.89 (7.81, 10.09)	9.53 (8.42, 10.75)	9.1 (8.02, 10.29)	9.22 (8.13, 10.43)	10.79 (9.59, 12.09)	12.66 (11.35, 14.08)	0.15
% ED presentations	1.5 (1.31, 1.67)	1.2 (1.04, 1.36)	1.4 (1.26, 1.61)	1.6 (1.42, 1.79)	1.6 (1.43, 1.8)	1.5 (1.37, 1.74)	1.4 (1.21, 1.57)	1.4 (1.27, 1.63)	1.4 (1.28, 1.64)	1.4 (1.24, 1.6)	1.4 (1.24, 1.57)	1.6 (1.41, 1.75)	0.472

Abbreviations: ED, emergency department, FB, foreign body, IR, incidence rate.

## Data Availability

The data presented in this study are available on request from the corresponding author. The data are not publicly available due to the privacy policy.

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
