# Peer review of "Epidemiology and Trends over Time of Foreign Body Injuries in the Pediatric Emergency Department"

_children, 2021, doi:10.3390/children8100938_

Round 1

Reviewer 1 Report

The current study has some certain values on public health and healthcare management regarding FB injuries in Italy. However, the study’s background was not framed properly, so it could not help rationalize the study’s objectives. Also, the study’s objectives were not clearly stated and somewhat different from what was shown in the Results. The methodology described was ambiguous, and the discussion lacked depth. Especially, the writing quality was very poor, which greatly undermined the clarity and understandability of the paper. 

Was the study’s abstract adequately written?

  • Please be more specific about the samples’ country.
  • What do the authors mean by “physiological cavities”?

Were the study’s objectives clearly stated?

  • The study’s objectives were not clearly stated and not rationalized adequately.

Were the research background and issues sufficiently framed?

  • The research background was weakly framed. There needs to be a better clarification on the FB situations in Italy and how healthcare systems in Italy have been operated to treat and reduce FB injuries.
  • The authors need to revise the Introduction substantially. It was superficially written, so it undermined the literature accuracy and clarity as well as the logical flow. Some examples are as follows:
    • “In Italy the estimated incidence of FB injuries in 2004 was approximately 0.2% [2]” -> please extend this statement with more details.
    • “Choking injuries related to inhalation of FBs are those that more frequently result in death” -> more than what?
    • “Understanding the epidemiological characteristics of the phenomenon is crucial to prevent FB injuries in children and to improve the patient’s management” -> how can understanding the epidemiological characteristics of the phenomenon help prevent FB and improve management? Please provide more details.
    • A lack of epidemiological knowledge on the FB injury problem is evidenced in the literature [19]. This gap possibly contributes to the low awareness of the problem among caregivers and childcare providers.” -> how can a lack of epidemiological knowledge lead to low awareness among caregivers and childcare providers?

Was the methodology well designed and implemented?

  • Lines 130-133 should be moved to the Methodology
  • What is the ICD strategy?
  • “This search strategy resulted more sensitive than the ICD-10 strategy.” -> what do the authors mean by “sensitive”?
  • Lines 111-114: what were the binary and multinomial logistic regressions used for?

Were the results clearly displayed and explained?

  • Lines 134-150 should be split into smaller paragraphs for easing readership.
  • Table 1 should be separated into two tables. One for comparing the differences between patients with FB injuries and patients with other diseases, one for comparing the injuries’ characteristics among patients with FB injuries.

Were the research findings, limitations, and recommendations for future research soundly and sufficiently discussed?

  • Lines 219-227: what can be drawn from contrasting the current study’ findings with results found in other countries
  • Lines 230-232: the statement is confusing
  • The originality or values of the current study were not clearly addressed in the Discussion. The authors have to indicate how their study can contribute in two ways. Practically, how can their findings be used to improve public health and healthcare management regarding FB injuries in Italy? Academically, what are their study’s novelty compared to previous studies (e.g. methodology, data, findings, etc.)?
  • The data used in the current study are not representative of FB injuries in Italy, so it needs to be included in the limitations with some notions on the proper usage of the current findings in policymaking.

Author Response

Thank you for giving us the opportunity to submit a revised draft of the manuscript “Epidemiology and trend over time of foreign body injuries in the Peadiatric Emergency Department” for publication in the Journal Children. We appreciate the time and effort that the reviewers dedicated to providing feedback on our manuscript and are grateful for the insightful comments on and valuable improvements to our paper. We have incorporated most of the suggestions made by the reviewers. Those changes are highlighted within the manuscript using the “Track Changes” function. Please see below, in Italic, for a point-by-point response to the reviewers’ comments and concerns. All page numbers refer to the revised manuscript file with tracked changes.

Reviewer 1

Open Review

English language and style

(x) Extensive editing of English language and style required
( ) Moderate English changes required
( ) English language and style are fine/minor spell check required
( ) I don't feel qualified to judge about the English language and style

Yes

Can be improved

Must be improved

Not applicable

Does the introduction provide sufficient background and include all relevant references?

( )

( )

( )

(x)

Is the research design appropriate?

( )

( )

(x)

( )

Are the methods adequately described?

( )

( )

(x)

( )

Are the results clearly presented?

( )

(x)

( )

( )

Are the conclusions supported by the results?

( )

( )

( )

(x)

Comments and Suggestions for Authors

The current study has some certain values on public health and healthcare management regarding FB injuries in Italy. However, the study’s background was not framed properly, so it could not help rationalize the study’s objectives. Also, the study’s objectives were not clearly stated and somewhat different from what was shown in the Results. The methodology described was ambiguous, and the discussion lacked depth. Especially, the writing quality was very poor, which greatly undermined the clarity and understandability of the paper. 

Thanks for providing feedback on our manuscript. With respect to the English language editing, we have sent the manuscript to an English editing service.

Was the study’s abstract adequately written?

  • Please be more specific about the samples’ country.

We added information on the sample’s country as suggest (line 16).

  • What do the authors mean by “physiological cavities”?

By physiological cavities we ment ears, nose, throat, gastrointestinal tract and eyes. We have now better clarified this in the text (lines 21-22)

Were the study’s objectives clearly stated?

The study’s objectives were not clearly stated and not rationalized adequately.

Thanks for pointing this out. We have now modified this section to clarify the definition of the study objectives (lines 182-189).

Were the research background and issues sufficiently framed?

  • The research background was weakly framed. There needs to be a better clarification on the FB situations in Italy and how healthcare systems in Italy have been operated to treat and reduce FB injuries.

Thanks for pointing this out. We modified the background to add information on national Italian suriveillance efforts on FB injuries monitoring and prevention.

  • The authors need to revise the Introduction substantially. It was superficially written, so it undermined the literature accuracy and clarity as well as the logical flow.

Thanks for this suggestion, we have now substantially revised the introduction accordingly.

Some examples are as follows:

  • “In Italy the estimated incidence of FB injuries in 2004 was approximately 0.2% [2]” -> please extend this statement with more details.

We have added more details on this point as suggested (lines 36-158).

  • “Choking injuries related to inhalation of FBs are those that more frequently result in death” -> more than what?

We have clarified this point (lines 162-163).

  • “Understanding the epidemiological characteristics of the phenomenon is crucial to prevent FB injuries in children and to improve the patient’s management” -> how can understanding the epidemiological characteristics of the phenomenon help prevent FB and improve management? Please provide more details.

Thanks for the suggestion, we have modified this sentence to better explained the message we were trying to convey (lines 172-174).

  • A lack of epidemiological knowledge on the FB injury problem is evidenced in the literature [19]. This gap possibly contributes to the low awareness of the problem among caregivers and childcare providers.” -> how can a lack of epidemiological knowledge lead to low awareness among caregivers and childcare providers?

We have endeavoured to better clarify this point (lines 172-179).

Was the methodology well designed and implemented?

  • Lines 130-133 should be moved to the Methodology

Thanks for the suggestion, we have made this change.

  • What is the ICD strategy?

Thanks for pointing this out. We have not specified it in lines 201-206.

  • “This search strategy resulted more sensitive than the ICD-10 strategy.” -> what do the authors mean by “sensitive”?

Thanks for pointing this out, better explained in lines 401-403.

  • Lines 111-114: what were the binary and multinomial logistic regressions used for?

We agree with the reviewer, the method section has been modified accordingly.

Were the results clearly displayed and explained?

  • Lines 134-150 should be split into smaller paragraphs for easing readership.

Thanks for the suggestion, the paragraph was split in smaller ones.

  • Table 1 should be separated into two tables. One for comparing the differences between patients with FB injuries and patients with other diseases, one for comparing the injuries’ characteristics among patients with FB injuries.

We agree with the reviewer; the table has been divided; another descriptive table reporting the characteristics of FB injury presentations has been included in the supplementary material.

Were the research findings, limitations, and recommendations for future research soundly and sufficiently discussed?

  • Lines 219-227: what can be drawn from contrasting the current study’ findings with results found in other countries

We have revised this part (lines 1263-1273)

  • Lines 230-232: the statement is confusing

Thanks for pointing this out, revised.

  • The originality or values of the current study were not clearly addressed in the Discussion. The authors have to indicate how their study can contribute in two ways. Practically, how can their findings be used to improve public health and healthcare management regarding FB injuries in Italy? Academically, what are their study’s novelty compared to previous studies (e.g. methodology, data, findings, etc.)?

Thanks for pointing this out, revised the discussion.

  • The data used in the current study are not representative of FB injuries in Italy, so it needs to be included in the limitations with some notions on the proper usage of the current findings in policymaking.

Thanks for the suggestion, added in lines 1547-1558.

Reviewer 2 Report

The authors evaluated the incidence rate of foreign body related injuries in children presenting to a tertiary - care paediatric emergency department in North Eastern Italy, to describe the characteristics of presentations compared with non- foreign body related injuries presentations and to evaluate the trend of the phenomenon over an eleven - year period. They concluded that the foreign body injury presentations is still increasing despite a growing awareness of policymakers on the topic. However, foreign body episodes that required hospitalization are reduced compared to previous data on the national territory. Moreover, most of the episodes at triage are coded as not dangerous for life.

I read the study with a great interest. This is well - designed and well - written study with an appropriate interpretation of the results. I have no major objections, only several minor objections / remarks for improvement:

  1. Abbreviation ‘’FB’’ has not been explained in the abstract. Please revise!
  2. Flow chart – penultimate box: the number is not visible (it was cropped). Please revise.
  3. Hypothesis as well as primary / secondary outcomes of the study are not clearly stated in methodology. Please provide a separate paragraph in methodology with hypothesis, primary and secondary outcome measurements of this study!
  4. Statistical analysis - Which software was used for statistical analysis? Please clarify. Which value represents statistical significance, it should be clearly stated.
  5. Presentation of the results – Please replace ‘’p’’ instead of ‘’p - value’’.
  6. Each abbreviation used in Tables should be explained in the legend of the Table (e. g. FB, N…)
  7. Table 3 is reader unfriendly, it is very hard to follow. It should be shortened and simplified. If it not possible to shorten, I vould advise to subdivide this Table in several smaller Tables such as 3a, 3b, 3c…
  8. Please include retrospective and single – center design of the study under the limitations of the study.

Author Response

Thank you for giving us the opportunity to submit a revised draft of the manuscript “Epidemiology and trend over time of foreign body injuries in the Peadiatric Emergency Department” for publication in the Journal Children. We appreciate the time and effort that the reviewers dedicated to providing feedback on our manuscript and are grateful for the insightful comments on and valuable improvements to our paper. We have incorporated most of the suggestions made by the reviewers. Those changes are highlighted within the manuscript using the “Track Changes” function. Please see below, in Italic, for a point-by-point response to the reviewers’ comments and concerns. All page numbers refer to the revised manuscript file with tracked changes.

Reviewer 2

Open Review

English language and style

( ) Extensive editing of English language and style required
( ) Moderate English changes required
(x) English language and style are fine/minor spell check required
( ) I don't feel qualified to judge about the English language and style

Yes

Can be improved

Must be improved

Not applicable

Does the introduction provide sufficient background and include all relevant references?

(x)

( )

( )

( )

Is the research design appropriate?

(x)

( )

( )

( )

Are the methods adequately described?

( )

(x)

( )

( )

Are the results clearly presented?

( )

(x)

( )

( )

Are the conclusions supported by the results?

(x)

( )

( )

( )

Comments and Suggestions for Authors

The authors evaluated the incidence rate of foreign body related injuries in children presenting to a tertiary - care paediatric emergency department in North Eastern Italy, to describe the characteristics of presentations compared with non- foreign body related injuries presentations and to evaluate the trend of the phenomenon over an eleven - year period. They concluded that the foreign body injury presentations is still increasing despite a growing awareness of policymakers on the topic. However, foreign body episodes that required hospitalization are reduced compared to previous data on the national territory. Moreover, most of the episodes at triage are coded as not dangerous for life.

Thanks for this summary and for providing constructive feedback on our manuscript.

I read the study with a great interest. This is well - designed and well - written study with an appropriate interpretation of the results. I have no major objections, only several minor objections / remarks for improvement:

  1. Abbreviation ‘’FB’’ has not been explained in the abstract. Please revise!
  1. Thanks, modified as suggest.

  1. Flow chart – penultimate box: the number is not visible (it was cropped). Please revise.
  1. Thanks, replace with the correct image.
  1. Hypothesis as well as primary / secondary outcomes of the study are not clearly stated in methodology. Please provide a separate paragraph in methodology with hypothesis, primary and secondary outcome measurements of this study!

Ho incluso una frase sulle ipotesi, vedete cosa vi sembra. Per gli outcomes aggiungerei un paragrafo in cui primary outcome e’ l’annual incidence rate e come viene definite/calcolato; come secondary outcomes metterei le caratteristche delle presentations con le specifiche variabili che sono state analizzate

Modified in the statistical methods section

  1. Statistical analysis - Which software was used for statistical analysis? Please clarify. Which value represents statistical significance, it should be clearly stated.

Thanks, for pointing this out. We have used the program R.

  1. Presentation of the results – Please replace ‘’p’’ instead of ‘’p - value’’.

We have made the suggested change.

  1. Each abbreviation used in Tables should be explained in the legend of the Table (e. g. FB, N…)

Thank you we have checked and ensured that abbreviations in the Tables are explained in the Table legend.

  1. Table 3 is reader unfriendly, it is very hard to follow. It should be shortened and simplified. If it not possible to shorten, I vould advise to subdivide this Table in several smaller Tables such as 3a, 3b, 3c…

Thank you for this suggestion, we have now modified Table 3 to enhance its readability.

  1. Please include retrospective and single – center design of the study under the limitations of the study.

We have added this limitation to the study limitation section (lines 313-314).

Round 2

Reviewer 1 Report

The paper has been improved significantly, especially the writing. However, several problems still have to be addressed before the paper is ready for publication.

Lines 110-113: it is not necessary to include a new subsection just to explain this study is exploratory. I recommend moving them to the end of the Introduction.

Lines 145-147: “The International Classification of Disease (ICD-10) is usually used for classifying the anatomical location of FB injuries. We preferred a search strategy that resulted in higher sensitivity than the ICD-10 strategy.” It is unclear whether the authors employed the ICD-10 to classify the anatomical location of FB injuries or used a different classification method. If the authors used another classification method, please specify. The statements also need rewriting for clarity.

Lines 390-395: The statements seem to be conflicting. What is the relation between the increase of choking incidents and the decrease in hospitalizations?

In addition, the paper's writing has to be carefully checked again because confusing expressions and grammatical errors still exist. I recommend turning off the track change function and using the highlight function instead. The track change function can sometimes result in errors.

Author Response

Comments and Suggestions for Authors

The paper has been improved significantly, especially the writing. However, several problems still have to be addressed before the paper is ready for publication.

Lines 110-113: it is not necessary to include a new subsection just to explain this study is exploratory. I recommend moving them to the end of the Introduction.

Thanks for pointing this out, modified as suggested.

Lines 145-147: “The International Classification of Disease (ICD-10) is usually used for classifying the anatomical location of FB injuries. We preferred a search strategy that resulted in higher sensitivity than the ICD-10 strategy.” It is unclear whether the authors employed the ICD-10 to classify the anatomical location of FB injuries or used a different classification method. If the authors used another classification method, please specify. The statements also need rewriting for clarity.

Thanks, we have amended the manuscript to clarify this sentence.

Lines 390-395: The statements seem to be conflicting. What is the relation between the increase of choking incidents and the decrease in hospitalizations?

In our results showed an increasing trend for choking injury-related presentations to the emergency department. However, only a small percentage (6%) of these presentations required hospitalization, in line with the national trends.. We endeavoured to better clarify this paragraph in the manuscript

In addition, the paper's writing has to be carefully checked again because confusing expressions and grammatical errors still exist. I recommend turning off the track change function and using the highlight function instead. The track change function can sometimes result in errors.

Thanks for this suggestion.

Round 3

Reviewer 1 Report

I think the manuscript is qualified for publication after English and style proofread.